# Antitumor Activity of USP7 Inhibitor GNE-6776 in Non-Small Cell Lung Cancer Involves Regulation of Epithelial-Mesenchymal Transition, Cell Cycle, Wnt/β-Catenin, and PI3K/AKT/mTOR Pathways

**DOI:** 10.3390/ph18020245

**Published:** 2025-02-12

**Authors:** Lipeng Wu, Long Lin, Meng Yu, Huajian Li, Yuanye Dang, Huosheng Liang, Guangyang Chen, Halimulati Muhetaer, Guodong Zheng, Jingjing Li, Xuejing Jia, Bo Wu, Chuwen Li

**Affiliations:** 1Guangzhou Municipal and Guangdong Provincial Key Laboratory of Molecular Target & Clinical Pharmacology, the NMPA and State Key Laboratory of Respiratory Disease, School of Pharmaceutical Sciences, Guangzhou Medical University, Guangzhou 511436, China; 2Phase I Clinical Trial Center, Guangzhou Eighth People’s Hospital, Guangzhou Medical University, Guangzhou 510440, China; 3School of Biomedical and Pharmaceutical Sciences, Guangdong University of Technology, Guangzhou 510006, China; 4Department of Rehabilitation Sciences, Faculty of Health and Social Sciences, Hong Kong Polytechnic University, Hong Kong SAR 999077, China; 5College of Food Science and Technology, Guangdong Ocean University, Zhanjiang 524088, China

**Keywords:** non-small cell lung cancer (NSCLC), USP7 inhibitor, GNE-6776, cell cycle, EMT

## Abstract

**Objective:** Non-small cell lung cancer (NSCLC) is a major cause of cancer-related deaths worldwide. This study investigated the effects and mechanisms of the USP7 inhibitor GNE-6776 on human NSCLC A549 and H1299 cells, providing insights for anti-NSCLC drug development. **Methods:** USP7 expression was analyzed in lung cancer tissue using data from public databases. RNA sequencing and functional enrichment analyses were conducted to explore differentially expressed genes (DEGs) and potentially related pathways. A549 and H1299 cells were treated with GNE-6776 at different concentrations, and its effects on cell proliferation, migration, invasion, apoptosis, mitochondrial membrane potential, and cell cycle were evaluated. Changes in protein expression following GNE-6776 treatment were assessed by Western blot. A xenograft tumor model in nude mice was used to evaluate the in vivo effects of GNE-6776. **Results:** GNE-6776 inhibited the proliferation, migration, and invasion of A549 and H1299 cells, induced apoptosis, and caused cells to arrest in the G1 phase in a concentration-dependent manner. GNE-6776 decreased the mitochondrial membrane potential, suppressed epithelial-mesenchymal transition (EMT) markers, and downregulated the PI3K/AKT/mTOR and Wnt/β-catenin signaling pathways. GNE-6776 significantly inhibited tumor growth without affecting body weight, reduced expression of CDK6, C-myc, and N-cadherin, and increased GSK3β expression in tumor tissue. **Conclusions:** In summary, GNE-6776 demonstrated potent anti-tumor activity in NSCLC both in vitro and in vivo. GNE-6776 suppresses NSCLC cell proliferation, invasion, and migration while promoting apoptosis by inhibiting the EMT and modulating the PI3K/AKT/mTOR and Wnt/β-catenin pathways. These findings support its potential as a therapeutic agent for treating NSCLC.

## 1. Introduction

Non-small cell lung cancer (NSCLC) accounts for 85% of all lung cancers and is the leading cause of cancer-related death worldwide [1,2,3,4]. The development of NSCLC involves multiple signaling pathways [5], such as the MAPK [6], PI3K/AKT/mTOR [7], Wnt [8], cell cycle [9], and DNA damage pathways [10]. Abnormal activation of the MAPK [11,12], PI3K/AKT/mTOR [13,14,15], and Wnt [16,17] pathways is frequently observed in NSCLC, and the activation of these pathways promotes the progression, drug resistance, proliferation, migration, and invasion of NSCLC. In recent years, an increasing number of clinical studies have demonstrated that epidermal growth factor receptor tyrosine kinase inhibitors (EGFR-TKIs) can prolong progression-free survival (PFS) and alleviate clinical symptoms of patients with advanced NSCLC harboring EGFR-sensitive mutations [18,19,20]. However, previous studies have found that despite the initial efficacy of EGFR-TKIs in treating these patients, their PFS rarely exceeded 14 months, and most patients developed resistance to the drugs over time [21,22]. Therefore, to further improve the clinical treatment outcomes of NSCLC, there is an urgent need to explore new therapeutic approaches.

Deubiquitinases (DUBs) have a crucial role in physiological processes, including cell differentiation, organelle biosynthesis, apoptosis, DNA repair, new protein synthesis, regulation of cell proliferation, protein transport, immune responses, and stress responses [23]. DUBs are categorized into seven primary subfamilies, which are distinguished by their abundance. These subfamilies encompass Ubiquitin-Specific Proteases (USP), Ovarian Tumor Proteases (OTU), Ubiquitin C-terminal Hydrolases (UCHs), JAB1/MPN/MOV34 metalloenzymes (JAMM), Machado-Josephin Domain-containing proteins (MJDs), MINDY proteins, and ZUFSP proteins [24]. Among them, the USP family is the largest and most extensively studied category within the DUB family [25]. Studies have found that multiple USP members are closely related to tumor metastasis and proliferation, including ubiquitin-specific protease 7 (USP7) [26,27]. USP7 promotes the stability of MDM2, thus maintaining the key tumor suppressor protein p53 at low levels [28,29]. Inhibition of USP7 caused DNA damage in chronic lymphocytic leukemia cells, leading to tumor cell death [30]. Furthermore, research suggests that overexpression of USP7 in human tumors inhibited tumor cell cycle arrest and promoted apoptosis, thereby promoting tumor progression. In lung cancer, one study indicated that USP7 was overexpressed in lung cancer, and this overexpression was directly related to the initiation of tumors and strongly correlated with a poor outcome in cancer patients [31]. USP7 was also shown to stabilize HK2 protein, therefore promoting NSCLC tumor progression [32]. In addition, we also verified from the DepMap, HPA, and UCSC Xena databases that Usp7 is a driver of NSCLC oncogenesis. (Appendix A). USP7 is increasingly gaining attention due to its potential as an important target in cancer therapy [33].

In recent years, USP7 inhibitors have attracted considerable attention in cancer therapy owing to the essential function of USP7. Previous research indicated that the small molecule HBX 41108 modulates p53 expression by blocking USP7’s de-ubiquitination activity, thereby inhibiting cancer cell growth [34]. The USP7 inhibitor P22077 exhibited inhibitory activity against cancer cell lines at low doses, operating through both p53-dependent and p53-independent pathways [35]. Recently, a novel small molecule, GNE-6776, was developed to specifically inhibit the function of USP7. Research has indicated that GNE-6776 suppression of USP7 was associated with increased p53 protein levels and apoptosis induction in cells infected with EBNA1 [36]. However, the therapeutic potential of GNE-6776 for the treatment of NSCLC remains unknown.

The article presents several innovative aspects in exploring the role and mechanisms of GNE-6776 in NSCLC. Firstly, it investigates GNE-6776 as a novel therapeutic agent for NSCLC, a field that has not been previously reported. Secondly, it elucidates the multi-pathway regulatory mechanisms of GNE-6776, including its modulation of the PI3K/AKT/mTOR and Wnt/β-catenin signaling pathways, which provides new insights into its anticancer effects. Thirdly, the study reveals that GNE-6776 inhibits the EMT process, a crucial mechanism for tumor metastasis and invasion. Furthermore, it demonstrates that GNE-6776 arrests NSCLC cells in the G1 phase, thereby suppressing cell proliferation. Additionally, the study employs both in vitro and in vivo models to comprehensively validate the efficacy of GNE-6776. Lastly, the findings support the potential clinical application of GNE-6776 as a novel therapeutic option for NSCLC, addressing the limitations of current targeted therapies, such as EGFR-TKIs.

## 2. Results

### 2.1. GNE-6776 Inhibits the Proliferation and the Clonogenic Ability of A549 and H1299 Cells

After seeding the cells into a 96-well plate, varying concentrations of GNE-6776 (0, 6.25, 25, and 100 µM) were introduced. Subsequent to a 24 or 48 h treatment period, CCK-8 solution was applied to assess cell viability. Compared to the control group, the viability of A549 and H1299 cells in the experimental groups treated with GNE-6776 exhibited a decrease, which was both concentration- and time-dependent (Figure 1A). Furthermore, the outcomes of a clone formation assay revealed that GNE-6776 markedly suppressed the colony-forming capacity of A549 and H1299 cells, with this inhibition occurring in a dose-dependent manner (Figure 1B).

Furthermore, we treated normal human lung epithelial cells Beas2B with GNE-6776, and the results indicated that even at a concentration as high as 100 µM, GNE-6776 had virtually no effect on Beas2B cells. This finding strongly demonstrates the high safety of GNE-6776 for in vitro applications (Appendix A).

### 2.2. GNE-6776 Promotes Apoptosis and Reduces the Mitochondrial Membrane Potential in A549 and H1299 Cells

Normal viable cells are not stained by either annexin-V or PI. Flow cytometry analysis using annexin-V/PI double staining demonstrated that GNE-6776 promoted apoptosis in A549 and H1299 cells. As shown in Figure 2A, as the concentration of GNE-6776 increased, the proportion of cells in the Q3 and Q2 quadrants gradually increased (particularly in the Q3 quadrant), indicating that as the GNE-6776 concentration increased, the number of apoptotic cells also gradually increased. This suggested that GNE-6776 promoted apoptosis in A549 and H1299 cells. A reduction in mitochondrial membrane potential is a characteristic feature observed during the initial phases of cellular apoptosis. After treating A549 and H1299 cells with GNE-6776, we used a JC-1 assay kit and flow cytometry to analyze the mitochondrial membrane potential. The mitochondrial membrane potential decreased as the concentration of GNE-6776 increased, suggesting that GNE-6776 affected the mitochondrial membrane potential, thereby promoting apoptosis in A549 and H1299 cells (Figure 2B). These results were consistent with the findings from annexin-V/PI staining experiments.

### 2.3. GNE-6776 Arrests A549 and H1299 Cells in the G1 Phase

After incubating A549 and H1299 cells separately with GNE-6776 at various concentrations for 24 h, flow cytometry analysis was performed with PI staining. Based on the cell cycle distribution histograms, it was evident that GNE-6776 arrested A549 and H1299 cells in the G1 phase (Figure 3A). The protein expression levels of key proteins in the G1 phase, including CDK6, cyclin D_1_, C-myc, and p21, were also measured, and the results were consistent with the flow cytometry findings. CDK6, cyclin D_1_, and C-myc have crucial roles in the cell cycle, and their downregulated protein levels contributed to the G1 arrest observed in the flow cytometry analysis. Furthermore, expression of p21, an important CDK inhibitor (CDKI), was upregulated, which inhibited CDK6 expression and subsequently suppressed cell cycle progression (Figure 3B).

### 2.4. GNE-6776 Inhibits Migration and Invasion of A549 and H1299 Cells

A549 and H1299 cells were treated with GNE-6776 and observed under a microscope at 0, 12, and 24 h to assess cell migration. Following treatment, a reduction in cell migration was noted for both cell lines. As the concentration of GNE-6776 increased, the wound healing rates decreased progressively, suggesting a concentration-dependent inhibition of A549 and H1299 cell migration by GNE-6776 (Figure 4A). To further investigate the effects of GNE-6776 on the migratory and invasive properties of A549 and H1299 cells, we conducted Transwell migration and invasion assays. The results revealed that GNE-6776 inhibited the migration and invasion of both cell lines, with the inhibitory effects becoming more pronounced as the concentration of GNE-6776 increased (Figure 4B). MMPs are known to significantly influence the invasiveness of NSCLC [37,38]. Therefore, we conducted a Western blot to determine the protein expression of MMP2 and MMP3. Our analysis of the MMP family protein levels revealed downregulation of both MMP2 and MMP3 (Figure 4C), aligning with the results obtained from the scratch assays and Transwell assays.

### 2.5. Transcriptome Sequencing of NSCLC Cells in the Control Group, Low-Concentration GNE-6776 Group and High-Concentration GNE-6776 Group

#### 2.5.1. Differential Gene Expression Analysis Between Control and Experimental Groups

A differential gene expression analysis was conducted by comparing the sequencing results from the control group with the low-concentration and high-concentration groups. The results revealed that there were 1332 DEGs between the low-concentration group and the control group, including 821 upregulated genes and 511 downregulated genes (Figure 5A). Between the high-concentration group and the control group, 3222 DEGs were identified, comprising 1207 upregulated genes and 2015 downregulated genes (Figure 5B), with 894 DEGs that overlapped between the two comparison groups (Figure 5C).

#### 2.5.2. GO and KEGG Analysis of Common Differential Genes in Both Groups

The 894 DEGs were subjected to KEGG and GO enrichment analysis using the Metascape database. The GO enrichment analysis indicated that the enriched biological processes encompassed small molecule biosynthetic processes, purine ribonucleoside triphosphate biosynthetic processes, purine nucleoside triphosphate metabolic processes, ribonucleoside triphosphate metabolic processes, steroid metabolic regulatory processes, purine ribonucleoside triphosphate metabolic processes, RNA polymerase III transcription factor activity, and regulation of the TORC1 signaling pathway. The differential genes were primarily localized in cellular components, including the lytic vacuoles, lysosomes, organelle inner membranes, extracellular matrix, external encapsulating structures, basement membranes, delta DNA polymerase complexes, mitochondrial inner membranes, focal adhesions, and collagen-containing extracellular matrices. The enriched molecular functions included glycosaminoglycan binding, myosin binding, ATPase inhibitor activity, integrin binding, organic anion transmembrane transporter activity, GDP binding, GTPase binding, nucleotide transmembrane transporter activity, ATPase binding, and lipid transporter activity (Figure 5D).

The KEGG analysis revealed that the enriched biological processes included interaction with extracellular matrix receptors, ribosome function, sphingolipid metabolism, the mTOR signaling pathway, lysosome activity, thermogenesis, the insulin signaling pathway, the AMPK signaling pathway, the PI3K/AKT signaling pathway, nucleotide metabolism, and arginine and proline metabolism (Figure 5E,F). Notably, the PI3K/AKT/mTOR and Wnt/β-catenin pathways are known to be crucial in tumorigenesis.

### 2.6. GNE-6776 Down-Regulates Vimentin and N-Cadherin, but Up-Regulates the Epithelial Marker E-Cadherin in NSCLC Cells

In addition to the MMP2 and MMP3 members of the MMP family, we found that GNE-6776 also affected levels of key proteins of the EMT, including vimentin, N-cadherin, and E-cadherin. These three proteins are important markers of the transformation of epithelial cells into mesenchymal cells. The results indicated that GNE-6776 inhibited the levels of the mesenchymal markers vimentin and N-cadherin and significantly increased levels of the epithelial marker E-cadherin (Figure 6).

### 2.7. GNE-6776 Down-Regulates the PI3K/AKT/mTOR Pathway in NSCLC Cells

The protein levels of the key proteins p-AKT/AKT and p-mTOR/mTOR in the PI3K/AKT/mTOR pathway were analyzed using Western blot analysis. Both proteins showed significantly decreased levels (Figure 7). Therefore, GNE-6776 affected the proliferation, migration, and invasion of A549 and H1299 cells through the PI3K/AKT/mTOR pathway.

### 2.8. GNE-6776 Suppresses Wnt/β-Catenin Pathway in NSCLC Cells

Western blot analysis of the key proteins GSK3β and p-β-catenin/β-catenin in the Wnt/β-catenin pathway showed that GNE-6776 intervention increased the protein levels of GSK3β and the ratio of p-β-catenin/β-catenin (Figure 8), thereby suppressing A549 and H1299 cell proliferation.

### 2.9. GNE-6776 Suppresses Tumor Growth in an A549 Xenograft Mouse Model Without Causing Significant Body Weight Loss

A549 cells were subcutaneously injected into nude mice. Two days later, low and high doses of GNE-6776, as well as DDP (cisplatin, a commonly used clinical drug in the treatment of NSCLC, was used as a positive drug to evaluate the relative efficacy of GNE-6776), were administered as interventions. Drugs were given every other day, and the control group received injections of normal saline. As the duration of the drug intervention increased, the high-dose GNE-6776 group exhibited the slowest tumor growth rate (Figure 9A). The tumor weights of mice in the low- and high-dose GNE-6776 groups and the DDP group were significantly reduced compared to those in the control group (Figure 9B). Compared with the low-dose GNE-6776 group, both the high-dose GNE-6776 group and the DDP group exhibited smaller tumor volumes (Figure 9C), indicating that the therapeutic effect of GNE-6776 on subcutaneously implanted A549 cell tumors in nude mice was dose-dependent within a certain dosage range. Additionally, hematoxylin and eosin (HE) staining of tumor tissues demonstrated that with increasing concentrations, the tumor cells exhibited progressively sparser arrangement, a gradual reduction in the increased nuclear chromatin observed in controls, and a marked escalation in the degree of necrosis (Appendix A).

The body weight of the tumor-bearing mice was analyzed, which is an important indicator of drug safety. At the conclusion of the administration period, there were no notable differences in the body weight of the mice among the low and high concentration GNE-6776 groups, the DDP group, and the control group (Figure 9D).

Subsequently, tumor tissue proteins were extracted to detect the expression of the key EMT protein N-cadherin, the oncogenic protein C-myc, and the key G1 phase protein CDK6. The results revealed that as the drug concentration increased, N-cadherin, C-myc, and CDK6 expression was downregulated. Additionally, the expression of the Wnt/β-catenin pathway inhibitory protein GSK3β was significantly upregulated (Figure 10).

## 3. Discussion

This study elucidates the profound impact of the USP7 inhibitor GNE-6776 on non-small cell lung cancer (NSCLC), showcasing its potential as a multi-mechanistic therapeutic agent. By targeting both cellular signaling pathways and biological processes integral to tumor progression, GNE-6776 emerges as a novel and effective candidate for combating NSCLC.

In NSCLC, the PI3K/AKT/mTOR pathway is closely related to tumorigenesis and disease progression [39,40]. It accelerates cell cycle progression, inhibits cell apoptosis, and promotes tumor cell migration. Studies have shown that PI3K/AKT/m-TOR pathway activation increases MMP2 and MMP9 protein levels, thereby enhancing the metastatic ability of cells [41]. GNE-6776 was found to significantly downregulate key phosphorylated proteins (p-AKT and p-mTOR), disrupting this oncogenic pathway. This inhibition likely accounts for the observed reduction in NSCLC cell proliferation and survival rates. Such findings align with the broader context of targeting hyperactivated PI3K/AKT/mTOR signaling in various cancers [42,43], emphasizing GNE-6776’s therapeutic specificity and efficacy. The Wnt signaling pathway is a highly conserved signaling pathway in evolution that controls cell growth, differentiation, apoptosis, and self-renewal. In tumor development, this pathway is often abnormally activated and has been linked to various cancers with high incidences [44]. Studies have shown that activation of the Wnt pathway can regulate tumor proliferation, migration, and invasion [45]. Our study found that GNE-6776 interfered with the Wnt/β-catenin signaling pathway. It enhanced GSK3β expression while promoting β-catenin phosphorylation, effectively suppressing its downstream oncogenic activity. This dual modulation of two critical pathways underscores GNE-6776’s capacity to address the multifactorial nature of NSCLC progression.

EMT is a cornerstone of cancer metastasis, enabling epithelial tumor cells to acquire mesenchymal traits and enhanced migratory capacity [46,47]. GNE-6776 treatment significantly downregulated mesenchymal markers (N-cadherin and vimentin) while upregulating the epithelial marker E-cadherin, thereby impeding EMT progression. This mechanism contributes directly to the compound’s anti-metastatic properties, as further evidenced by reduced invasion and migration rates in A549 and H1299 cells.

The study demonstrated that GNE-6776 induces apoptosis in a dose-dependent manner, as evidenced by Annexin V/PI staining. This effect is complemented by a marked decrease in mitochondrial membrane potential, which is a hallmark event in the early stages of apoptosis, and once the mitochondrial membrane potential collapses, apoptosis becomes irreversible [48]. These findings suggest that mitochondrial dysfunction and subsequent apoptotic cascades are primary mechanisms by which GNE-6776 eliminates NSCLC cells.

The cell cycle is a fundamental process in cellular activities [49]. Precise regulation of this process is crucial for maintaining cell homeostasis, and its dysregulation can lead to uncontrolled cell proliferation and cancer formation [50,51]. Cyclins, CDKs, and CDKIs, which are pivotal in the cell cycle, functionally collaborate with and complement each other in regulating the cell cycle, growth, and development to achieve precise control of cell proliferation and individual development [52,53]. To analyze the effect of GNE-6776 on the cell cycle of A549 and H1299 cells, we performed flow cytometry analysis and found that GNE-6776 significantly arrested cells in the G1 phase of the cell cycle, correlating with a reduction in cyclin D1, CDK6, and C-myc protein levels. Upregulation of the CDK inhibitor p21 further reinforced this effect, highlighting GNE-6776’s ability to disrupt cell cycle progression. This mechanism is critical, as it directly impedes the proliferation of tumor cells.

The in vivo efficacy of GNE-6776 is particularly noteworthy given its ability to recapitulate the in vitro findings. The inhibition of tumor growth observed in mice treated with GNE-6776 is likely attributed to the compound’s ability to target the same pathways and processes that were identified in vitro. Specifically, the downregulation of N-cadherin, C-myc, and CDK6, as well as the upregulation of GSK3β, in tumor tissues treated with GNE-6776, aligns with the in vitro results showing that GNE-6776 suppresses the PI3K/AKT/mTOR and Wnt/β-catenin pathways, inhibits EMT, and arrests cells in the G1 phase of the cell cycle. Moreover, the in vivo model allowed us to assess the safety and tolerability of GNE-6776, which is crucial for its potential clinical translation. The lack of significant body weight loss in GNE-6776-treated mice indicates that the compound is well-tolerated and does not cause severe systemic toxicity. The close correlation between the effects observed in vitro and in vivo underscores the robustness of our findings and highlights the potential of GNE-6776 as a multifaceted therapeutic agent for treating NSCLC. The ability of GNE-6776 to inhibit tumor growth in vivo, while maintaining a good safety profile, further highlights its promising clinical prospects. Current NSCLC treatments, such as EGFR-TKIs, are limited by the rapid development of resistance, which often stems from secondary mutations or activation of alternative pathways [54,55]. GNE-6776, by simultaneously targeting multiple pathways and processes, addresses these limitations. This multi-targeted approach complementing existing therapies, such as EGFR-TKIs or immune checkpoint inhibitors not only enhances therapeutic efficacy but also reduces the likelihood of resistance development.

## 4. Materials and Methods

Here, we have presented a flowchart of the experimental procedure to facilitate a better understanding of the content of this experiment (as illustrated in Figure 11).

### 4.1. Cell Culture and Drug

Human NSCLC cell lines (A549 and H1299) and Beas2B were purchased from ATCC and cultured in RPMI 1640 medium (Gibco, Grand Island, NE, USA) supplemented with 10% fetal bovine serum (Vivacell, Shanghai, China) and 1% penicillin-streptomycin (Gibco, USA) at 37 °C in a 5% CO_2_ atmosphere. The cells were passaged regularly.

GNE-6776 (MCE, Monmouth Junction, NJ, USA) was dissolved in Dimethyl sulfoxide (DMSO) and its stock solution concentration was 100 mM.

### 4.2. RNA Sequencing, Differential Expression Analysis, and KEGG Analysis

A549 cells were divided into a control group, a low-dose GNE-6776 group (25 μM), and a high-dose group (100 μM), and RNA-seq was performed after 24 h of treatment. Total RNA was extracted from cells using TRIzol reagent (Ambion, Austin, TX, USA) according to the manufacturer’s instructions. Library construction and mRNA sequencing were performed by Majorbio Bio-Pharm Technology Co., Ltd. (Shanghai, China). mRNA was enriched using oligo (dT) beads and then fragmented. Reverse transcription was performed using random N6 primers to form double-stranded DNA, and the raw data underwent quality control using SOAPnuke software v2.1.0 (Shenzhen, China). Reference genome alignment was performed using HISAT software 2.2.1 (Baltimore, MD, USA) based on the Burrows–Wheeler transform and Ferragina–Manzini method. Finally, differential gene expression was analyzed using the DESeq2 package, with screening criteria of FDR < 0.05 and |log2 fold change| > 1. To explore the pharmacological mechanism of GNE-6776 against NSCLC, we identified common DEGs by taking the intersection of DEGs obtained from comparisons between the control group and the low-dose group, as well as between the control group and the high-dose group. The common DEGs were then subjected to KEGG and GO enrichment analyses using the Metascape database, with enrichment criteria set at *p* < 0.01 and a minimum overlap of 3.

### 4.3. CCK-8 Assay for Cell Viability

Cells (A549 and H1299) in the logarithmic growth phase were seeded into 96-well plates at 5000 cells per well. Following 24 h incubation, cells were treated with GNE-6776 at 0, 6.25, 25, or 100 µM and incubated for an additional 24 or 48 h. Absorbance at 450 nm was then measured using a microplate reader after adding 10% CCK-8 reagent and incubating for 2–3 h.

### 4.4. Clone Formation Assay

Cells (A549 and H1299) in the logarithmic growth phase were plated onto six-well plates at a density of 1000 cells per well. Following cell adherence, the cells were exposed to GNE-6776 at concentrations of 0, 25, and 100 μM. The cells were then cultivated for 10 days, with the medium being replaced every 3 days (while maintaining the presence of GNE-6776 throughout the 10-day period to ensure continuous treatment). After the cultivation period ended, the cells were rinsed with PBS, fixed using 4% paraformaldehyde for 30 min, stained with crystal violet for 10 min, rinsed again with PBS, and images were captured using a microscope after air-drying. Lastly, the images were subjected to analysis using ImageJ software 1.54d (National Institutes of Health, Bethesda, MD, USA).

### 4.5. Annexin V-FITC Apoptosis Assay for Cell Apoptosis Detection

Cells (A549 and H1299) from the control group, 25 μM drug treatment group, and 100 μM drug treatment group were collected, resuspended in PBS, and mixed with 195 μL of loading buffer (Beyotime, Shanghai, China). Then, 10 μL of PI and 5 μL of FITC were added, and the cells were stained at room temperature for 10 min. The cells were analyzed using a flow cytometer, and the data were analyzed with FlowJo software 10.8.1.

### 4.6. JC-1 Assay for Mitochondrial Membrane Potential Assessment

Cells (A549 and H1299) from three groups—control, 25 μM drug-treated, and 100 μM drug-treated—were harvested. Subsequently, 0.5 mL of medium was added to each sample, followed by the addition of 0.5 mL of JC-1 staining working solution (Beyotime, China), and the mixtures were thoroughly mixed. The cells were then incubated at 37 °C for a duration of 30 min. Post-incubation, the cells were subjected to centrifugation, the supernatants were discarded, and the cells were washed twice using JC-1 buffer. Thereafter, 1 mL of cell medium was added to each sample. Flow cytometry was employed for cell analysis, and the obtained data were processed using FlowJo software.

### 4.7. Flow Cytometry with PI Staining for Cell Cycle Analysis

Cells (A549 and H1299) from the control group, as well as the 25 μM and 100 μM drug treatment groups, were harvested and fixed with pre-chilled 75% ethanol for a duration of 4 h. Subsequently, the cells were centrifuged at 4 °C, resuspended in PBS, centrifuged once more, and the supernatant was removed. Next, 0.5 mL of propidium iodide staining solution (Beyotime, China) was introduced, and the cells were incubated at 37 °C in darkness for 30 min. The cells were then subjected to analysis using a flow cytometer (Beckman Coulter, Brea, CA, USA), and the obtained data were processed using FlowJo software.

### 4.8. Wound Healing Assay

For the wound healing assay, six-well plates were pre-marked with parallel lines spaced 0.5 cm apart using a marker pen, with each well containing five such lines to delineate the observation zones. A549 and H1299 cells were plated at a density of 6 × 10^5^ cells per well. Once the cells attained 90% confluence, a scratch was created perpendicular to the marked lines using a 200-μL pipette tip. The cells were then rinsed three times with PBS. Subsequently, GNE-6776 was added at concentrations of either 25 μM or 100 μM. Microscopic images using a Leica microscope (Mannheim, Germany) were captured at 0, 12, and 24 h, and the results were subsequently analyzed using ImageJ software.

### 4.9. Transwell Migration Assay

Cells (A549 and H1299) were harvested and suspended in serum-free medium at a concentration of 3 × 10^5^ cells per well and then seeded into the upper chamber of a Transwell plate (Corning, NY, USA). Subsequently, 650 μL of complete medium containing 10% FBS was added to the lower chamber. GNE-6776 was introduced into the upper chamber to reach final concentrations of 25 and 100 μM. Following a 24 h incubation period, the supernatant was aspirated, and the cells were washed thrice with PBS before being fixed with 4% paraformaldehyde. Non-migrated cells in the upper chamber were then removed using a cotton swab. The cells were stained with crystal violet, allowed to air-dry, and imaged using a Leica microscope (Germany).

### 4.10. Transwell Invasion Assay

Diluted Matrigel (sourced from Corning, USA) was applied to the upper chamber of a Transwell plate and allowed to incubate for 3 h. Following incubation, the cells were harvested, suspended in serum-free medium at a density of 6 × 10^5^ cells per well, and then plated onto the upper chamber of the Transwell plate. The subsequent steps mirrored those of the migration assay.

### 4.11. Western Blot

After treating A549 and H1299 cells with GNE-6776, cellular proteins were extracted using RIPA lysis buffer (Beyotime, China). Protein quantification was carried out with a BCA protein concentration assay kit (Thermo Fisher, Waltham, MA, USA). The proteins (20 to 30 μg per lane) were then separated via electrophoresis and transferred onto PVDF membranes (Merck, Darmstadt, Germany) utilizing a wet transfer technique. Subsequently, the membranes were blocked with 5% skim milk (FD Bioscience, Hangzhou, China) for a duration of 1 h. Primary antibodies were introduced, and the membranes were incubated overnight at 4 °C with shaking. Following three washes of 5 min each, the membranes were incubated with the appropriate HRP-conjugated secondary antibodies (obtained from Affinity, Beachwood, OH, USA) for 1 h at 25 °C with shaking. After another set of three 5 min washes, an ECL luminescent substrate (from NCM Biotech, Suzhou, China) was applied for imaging and analysis using the ChemiDoc™ XRS+ imaging system (manufactured by Bio-Rad, Hercules, CA, USA). Antibodies to N-cadherin, GAPDH, β-tubulin, p-mTOR, mTOR, p-AKT, AKT, MMP2, MMP3, GSK3β, p-β-catenin, β-catenin, and cyclinD_1_ were from Affinity (USA); antibody to vimentin was from Servicebio; antibodies to p21, E-cadherin, CDK6, C-myc, and β-actin were from Proteintech (Rosemont, IL, USA). The dilution ratio for all primary antibodies was 1:1000, and the dilution ratio for all secondary antibodies was 1:3000.

### 4.12. Xenograft Mouse Model

Four-week-old male nude mice were purchased from the Shanghai Model Organisms Center. The initial weight of the mice was 19 ± 0.4 g. The mice were maintained in accordance with institutional guidelines under conditions approved by the Committee for Animal Experimentation of Guangzhou Medical University and the Animal Ethics Committee of Guangzhou Medical University (approval number: GY2024-370). Animals were kept at room temperature (18–22 °C) and humidity (40–60%). A549 cells were resuspended in PBS and adjusted to a concentration of 3 × 10^7^ cells/mL. Each mouse was subcutaneously inoculated with 100 μL of the cell suspension on the right dorsolateral region near the axilla. Subsequently, the mice were randomly divided into four groups: a control group (normal saline via intraperitoneal injection), a low-concentration group (15 mg/kg GNE-6776 via intraperitoneal injection), a high-concentration group (30 mg/kg GNE-6776 via intraperitoneal injection), and a cisplatin group (2 mg/kg DDP via intraperitoneal injection). The tumor volumes were calculated using the formula: tumor volume = (width^2^ × length)/2. One week after drug administration (drugs were given every other day), the nude mice were euthanized by cervical dislocation, and the transplanted tumor tissue was removed and weighed. Subsequently, total protein was extracted from the tumor tissue using RIPA lysis buffer. Western blotting was used to detect the protein expression in the transplanted tumors of N-cadherin, a key protein in the EMT; C-myc and CDK6, key proteins involved in the cell cycle; and GSK3β in the Wnt/β-catenin pathway.

### 4.13. Statistical Analysis

The experimental data were statistically evaluated utilizing GraphPad Prism 9 (Inc., San Diego, CA, USA). The results were presented as the mean ± standard deviation (SD). For comparisons between two groups, the unpaired two-tailed Student’s *t*-test was employed. When comparing data among multiple groups, one-way analysis of variance (ANOVA) was used, followed by Tukey’s post hoc test. Statistical significance was defined as *p* < 0.05.

## 5. Conclusions

In summary, our study highlights GNE-6776, a USP7 inhibitor, as a promising therapeutic for NSCLC. In vitro, GNE-6776 concentration-dependently inhibited NSCLC cell proliferation, migration, and invasion; induced apoptosis; and arrested the cell cycle. It also downregulated EMT markers, suggesting anti-metastatic effects, and modulated key cancer pathways, such as PI3K/AKT/mTOR and Wnt/β-catenin. In vivo, GNE-6776 dose-dependently suppressed tumor growth in a xenograft mouse model with good tolerability (as illustrated in Figure 12). Its multi-targeted approach holds potential for overcoming therapy resistance and improving NSCLC outcomes. Further development of GNE-6776 could revolutionize NSCLC treatment, benefiting patients globally.

## Figures and Tables

**Figure 1 pharmaceuticals-18-00245-f001:**
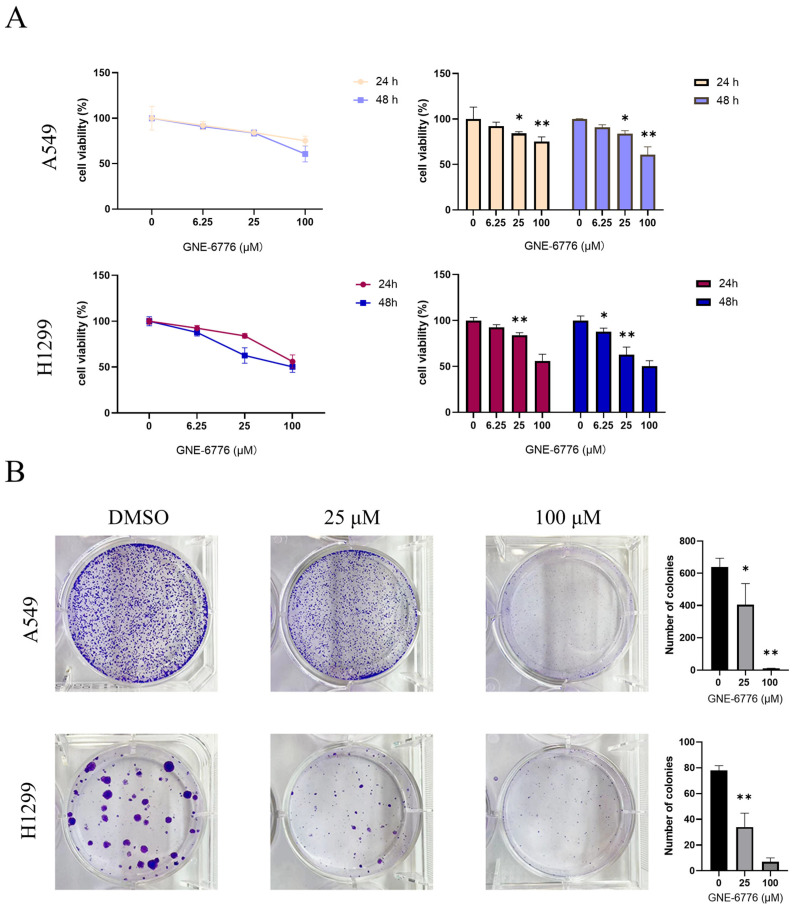
Effect of GNE-6776 on the proliferation of A549 and H1299 cells. Cells were exposed to GNE-6776 at increasing concentrations (0, 6.25, 25, and 100 μM) for 24 or 48 h. The viability of the cells was then determined using a CCK-8 assay (**A**). Additionally, representative images of A549 and H1299 cell colonies treated with various concentrations of GNE-6776 for 10 days are shown (**B**) with similar statistical analysis. The results are presented as the mean ± SD (*n* = 9), and statistical analysis was conducted using one-way ANOVA. * *p* < 0.05, ** *p* < 0.01, compared with the control group.

**Figure 2 pharmaceuticals-18-00245-f002:**
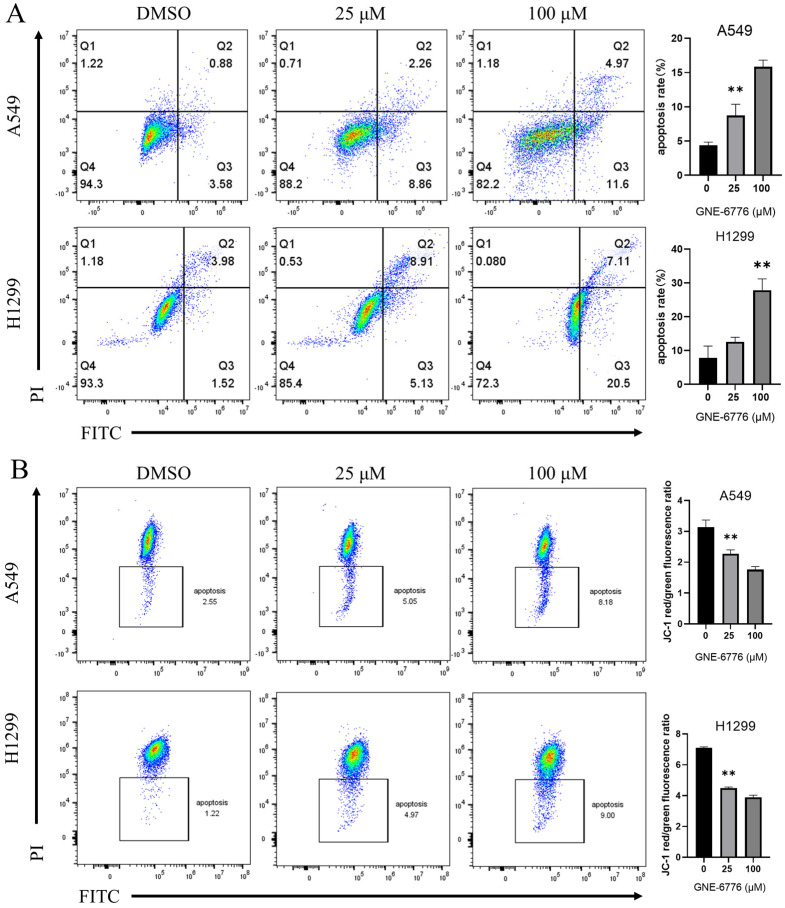
Effect of GNE-6776 on apoptosis of NSCLC cells. Flow cytometry analysis of apoptosis in A549 and H1299 cells treated with varying concentrations of GNE-6776 for 24 h. Cells were stained with (**A**) annexin V-FITC/PI and (**B**) JC-1 assay. Quantification of (**A**) apoptotic rate and (**B**) red/green fluorescence ratio were also shown. The results are presented as the mean ± SD (*n* = 9), and statistical analysis was conducted using one-way ANOVA. ** *p* < 0.01, compared with the control group.

**Figure 3 pharmaceuticals-18-00245-f003:**
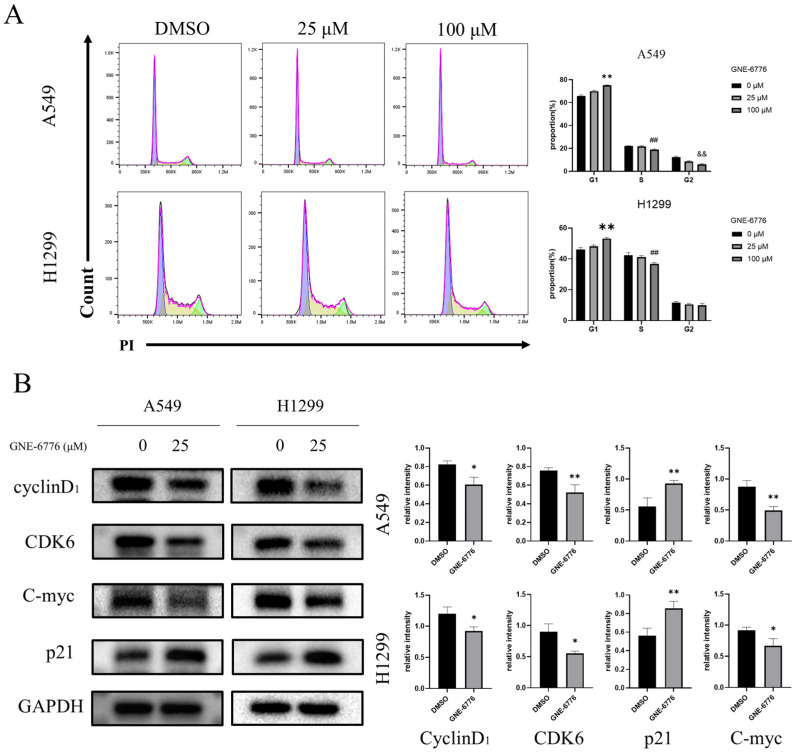
Effect of GNE-6776 on the cell cycle distribution and cell cycle-related protein expression of NSCLC cells. (**A**) Flow cytometry was used to analyze the cell cycle distribution of A549 and H1299 cells following a 24 h treatment with different concentrations of GNE-6776. The purple section represents the G1 phase, the yellow section represents the S phase, and the green section represents the G2 phase. The results are presented as the mean ± SD (*n* = 9), and statistical significance was determined using one-way ANOVA. ** indicates *p* < 0.01 for the G1 phase compared to the control group, ## indicates *p* < 0.01 for the S phase compared to the control group, and && indicates *p* < 0.01 for the G2 phase compared to the control group. (**B**) western blot analysis was performed to assess the expression of key G1 phase proteins (cyclin D_1_, CDK6, C-myc, and p21) in A549 and H1299 cells treated with 25 μM GNE-6776 for 24 h. GAPDH served as a loading control. The data are presented as the mean ± SD (*n* = 3), and statistical significance was analyzed using Student’s *t*-test. * *p* < 0.05, ** *p* < 0.01, compared to the control group.

**Figure 4 pharmaceuticals-18-00245-f004:**
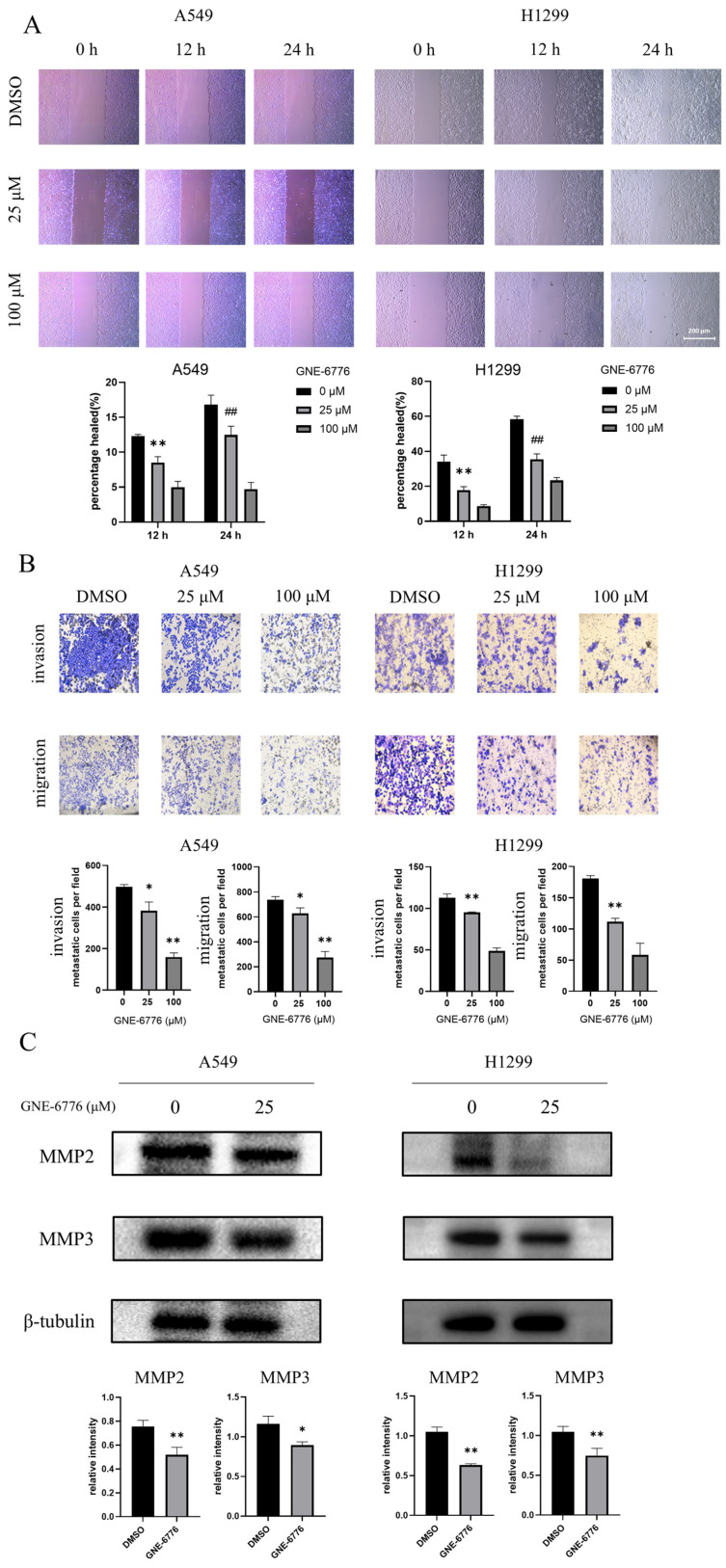
GNE-6776 inhibits the migration and invasion of NSCLC cells and the expression of MMPs. (**A**) Representative images show the wound healing of A549 and H1299 cells treated with different concentrations of GNE-6776 at 0, 12, and 24 h. Data are mean ± SD (*n* = 9), and statistical significance was determined by one-way ANOVA. ** represents *p* < 0.01 compared to the control group at the 12-h time point, ## represents *p* < 0.01 compared to the control group at the 24-h time point. (**B**) Representative images depict migrated (lower row) and invaded (upper row) A549 and H1299 cells following treatment with GNE-6776. Data are mean ± SD (*n* = 9), with statistical significance analyzed by one-way ANOVA. * *p* < 0.05, ** *p* < 0.01, compared to the control group. (**C**) Western blot analysis of MMPs in NSCLC cells treated with 25 μM GNE-6776 for 24 h is shown, with β-tubulin as a loading control. Data are mean ± SD (*n* = 3), and statistical significance was analyzed by Student’s *t*-test. * *p* < 0.05, ** *p* < 0.01, compared to the control group.

**Figure 5 pharmaceuticals-18-00245-f005:**
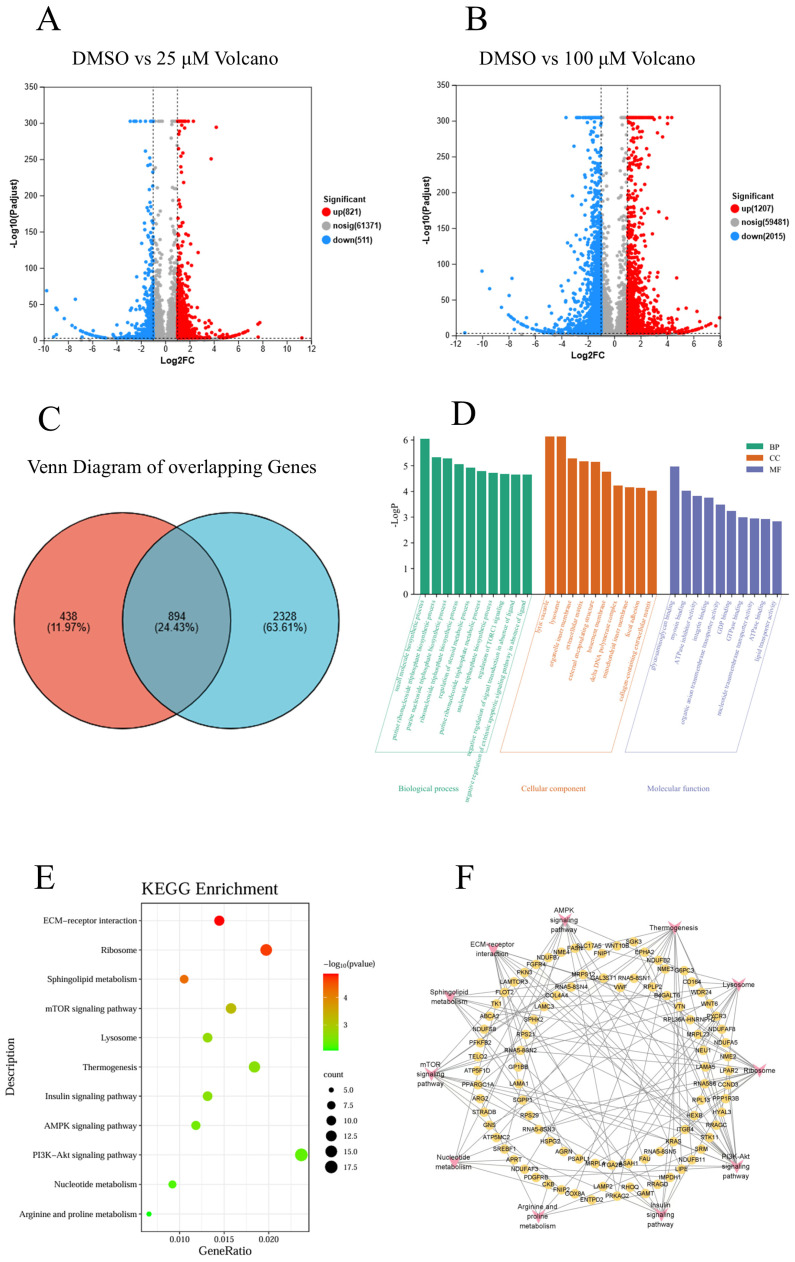
Differentially expressed genes (DEGs) in A549 cells treated with GNE-6776 are shown as volcano and Venn diagrams. KEGG pathway and GO enrichment analyses of DEGs in A549 cells treated with GNE-6776, and the network diagram of KEGG pathways and corresponding targets affected by GNE-6776 in NSCLC cells. The volcano plots (**A**,**B**) visualize the distribution of differentially expressed genes (DEGs) of the low-concentration treatment group and high-concentration treatment group compared to the control group, based on fold change and statistical significance. In the Venn diagram (**C**), the red circle represents DEGs of A549 cells treated with a low concentration of GNE-6776 compared to the control, and the blue circle represents DEGs of A549 cells treated with a high concentration of GNE-6776 compared to the control. The overlapping region shows genes shared across groups, indicating potential common targets or pathways modulated by GNE-6776 in NSCLC. (**D**–**F**) Shared DEGs were subjected to KEGG and GO enrichment analysis revealing a close association between GNE-6776 and PI3K/AKT/mTOR and Wnt/β-catenin pathways.

**Figure 6 pharmaceuticals-18-00245-f006:**
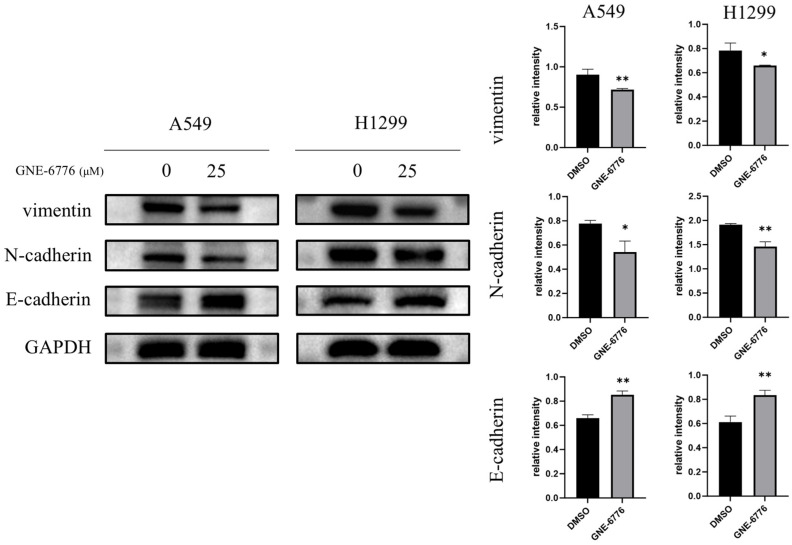
Effects of GNE-6776 on EMT protein expression in NSCLC cells. Western blot analysis revealed the expression levels of EMT markers (E-cadherin, N-cadherin, vimentin) in NSCLC cells after 24 h treatment with 25 μM GNE-6776. GAPDH served as a loading control. Data are mean ± SD (*n* = 3), and statistical significance was determined using Student’s *t*-test. * *p* < 0.05, ** *p* < 0.01, compared to the control group.

**Figure 7 pharmaceuticals-18-00245-f007:**
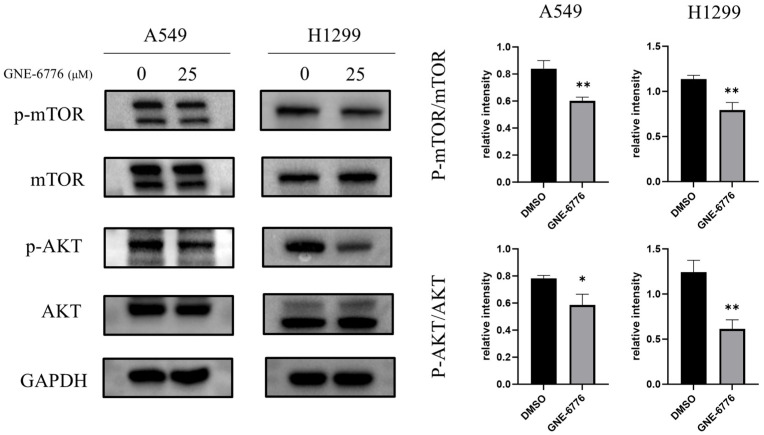
Effects of GNE-6776 on PI3K/AKT/mTOR signaling pathway protein expression in NSCLC cells. NSCLC cells treated with 25 μM GNE-6776 for 24 h were analyzed for levels of PI3K/AKT/mTOR signaling pathway proteins (PI3K, p-AKT/AKT, and p-mTOR/mTOR) using Western blot analysis. GAPDH was used as a loading control. Statistical significance was analyzed using Student’s *t* test. * *p* < 0.05, ** *p* < 0.01, compared to the control group.

**Figure 8 pharmaceuticals-18-00245-f008:**
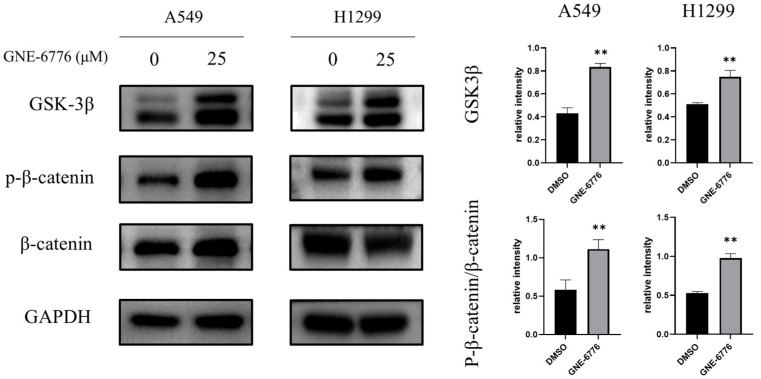
Effects of GNE-6776 on Wnt/β-catenin signaling pathway protein expression in NSCLC cells. NSCLC cells were treated with 25 μM GNE-6776 for 24 h. Levels of the Wnt/β-catenin signaling pathway proteins GSK3β and p-β-catenin/β-catenin were determined by Western blot analysis. GAPDH was used as a loading control. Statistical significance was analyzed using Student’s *t* test. ** *p* < 0.01, compared to the control group.

**Figure 9 pharmaceuticals-18-00245-f009:**
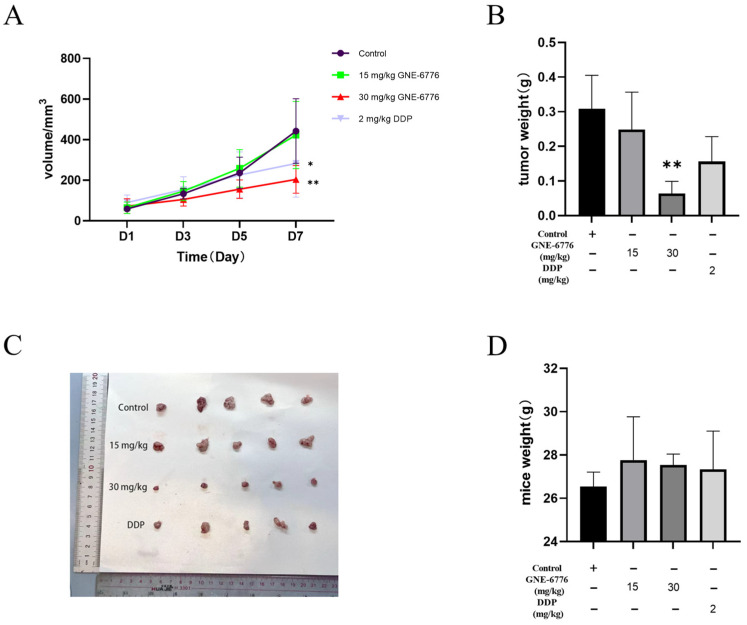
Effect of GNE-6776 on the Growth of Subcutaneously Implanted A549 Cell Tumors in Nude Mice. (**A**) Volumes, (**B**) weights, and (**C**) representative images of xenograft tumors excised from nude mice in each treatment group (vehicle control, low-dose GNE-6776 (15 mg/kg), high-dose GNE-6776 (30 mg/kg), DDP (2 mg/kg)) after 7 days along with (**D**) the corresponding body weights at the end of the treatment period. Data are presented as mean ± SD (*n* = 5). Statistical significance was analyzed using one-way ANOVA. * *p* < 0.05, ** *p* < 0.01, compared to the control group.

**Figure 10 pharmaceuticals-18-00245-f010:**
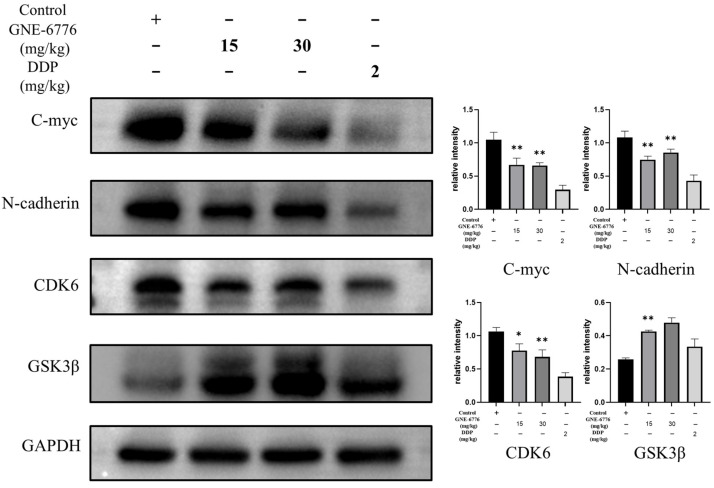
GNE-6776 regulates key signaling pathways in xenograft tumor tissue. Nude mice were treated with the vehicle control, GNE-6776 (15 and 30 mg/kg), or DDP, and a Western blot analysis of the protein levels in the tumor tissue was performed. Representative blots showed that the levels of an EMT marker (N-cadherin), cell proliferation marker (C-myc), and pivotal G1 phase marker (CDK6) decreased, and levels of a Wnt pathway-related marker (GSK3β) increased. GAPDH was used as a loading control. Data are expressed as mean ± SD (*n* = 4). Statistical significance was analyzed using one-way ANOVA. * *p* < 0.05, ** *p* < 0.01, compared to the control group.

**Figure 11 pharmaceuticals-18-00245-f011:**
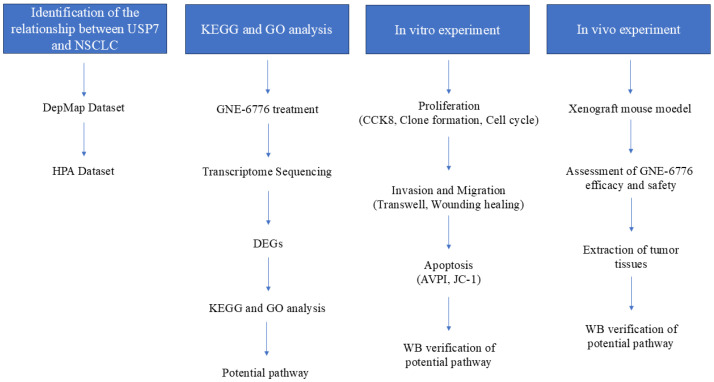
Illustration of the experimental procedure. This flowchart clearly presents an in-depth research process of GNE-6776 targeting NSCLC, aiming to comprehensively reveal the potential mechanism of action of GNE-6776 on NSCLC through a series of meticulously designed experiments and analyses. The entire research process is carefully divided into the following four core parts: “Identification of the relationship between USP7 and NSCLC”, “KEGG and GO analysis”, “In vitro experiment”, and “In vivo experiment”. Through the close connection and in-depth exploration of these four parts, this flowchart not only clearly demonstrates the research thread of GNE-6776’s impact on NSCLC but also highlights the core content and scientific value of the research.

**Figure 12 pharmaceuticals-18-00245-f012:**
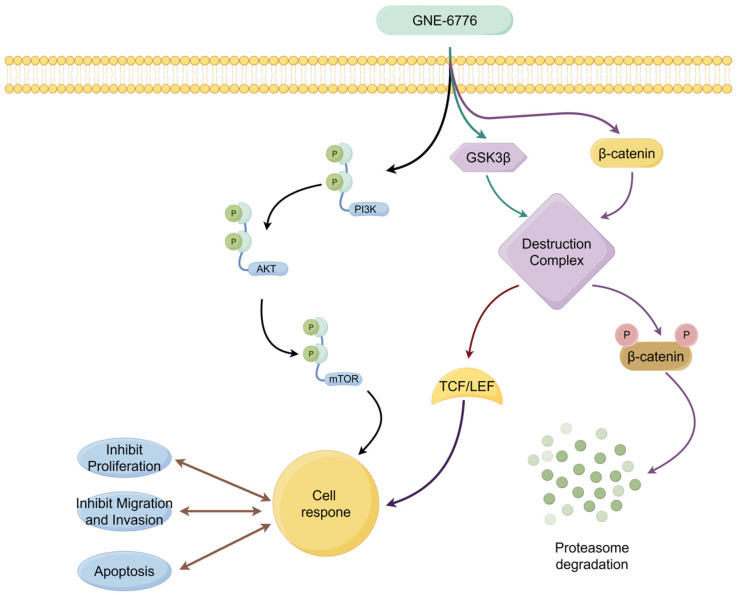
The mechanism of action of GNE-6776 in NSCLC cells. The mechanism of action of USP7 inhibitor GNE-6776 in the NSCLC cell signaling pathway. GNE-6776 influences cellular responses through multiple pathways (PI3K/AKT/mTOR and Wnt/β-catenin pathways), including inhibiting proliferation, migration, and invasion, as well as promoting apoptosis.

## Data Availability

All data associated with this study are available from the corresponding author upon reasonable request.

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
