# Peer review of "Antitumor Activity of USP7 Inhibitor GNE-6776 in Non-Small Cell Lung Cancer Involves Regulation of Epithelial-Mesenchymal Transition, Cell Cycle, Wnt/β-Catenin, and PI3K/AKT/mTOR Pathways"

_pharmaceuticals, 2025, doi:10.3390/ph18020245_

Round 1
Reviewer 1 Report
Comments and Suggestions for Authors
Article Review: USP7 inhibitor GNE-6776 represses the proliferation, invasion, and migration of non-small cell lung cancer by regulating the epithelial-mesenchymal transition and cell cycle
The article is presented in a clear and understandable way, however there are some suggestions and doubts about the information presented:
1. Italicize "in vivo" in the abstract, also in line 101.
2. Improve grammar on line 87-88.
3. Review and compare the results with the article https://doi.org/10.1016/j.celrep.2024.114917, since the inhibitor GNE-6776 is also evaluated on lung cancer cells.
4. Subscript the 2 of CO2. Line 400.
5. What solvent was used to dissolve GNE-6776?
6. In experiment 4.4, was the treatment only applied at the beginning of the 10 days? This was because the medium was changed and the GNE-6776 molecule was lost with this change.
7. How were the cells acquired? Donation, purchase from ATCC?
8. Missing full stop, line 443.
9. Line 480. What were the cells photographed with? Specify
10. Regarding the antibodies used, what was the brand of each one? It is not clear from the document.
11. How much protein was loaded in the western blot assay, was it the same amount for all assays, or was there some variation depending on the protein of interest.
12. In the animal model, did the control group receive vehicle administration? And if so, which one was used?
13. What post hoc test is used after performing ANOVA?
14. In methodology 4.3 CCK-8 Assay for Cell Viability, specify the concentrations of GNE-6776 used.
15. In Figure 1A, separate cell viability (%).
16. Line 121-122 repeats the information, modify.
17. Line 150-152 repeats the information, modify.
18. Why does Figure 3B only show the 25 micromolar concentration?
19. What is the amount of DMSO used in the control groups?
20. Figure 4c. Place the image in the same order as the graph, i.e. MMP2 first.
21. In Figure 4, the loading protein is not specified, in the other figures it is mentioned, to homogenize the information in the figures.
22. Figures 7 and 8, it remains to be mentioned what concerns statistical analysis.
23. In the in vivo model, only one dose was administered and at what point in the experiment was it administered? Mention in the methodology.
24. Figure 9 is missing a description, please add it.
25. Figure 9 and 10 have had their figure captions moved. Figure 10 is missing information on statistical analysis.
26. Line 373, remove double point.
27. The discussion lacks the issue of the in vivo model, how does it relate to the effects observed in vitro?
Author Response
Thank you very much for your attention and support. I have compiled the information you need. Please review the attachment at your convenience. If you have any questions or need further assistance, please feel free to contact me. I look forward to your reply. Thank you!

Reviewer 2 Report
Comments and Suggestions for Authors
In the paper "Antitumor activity of USP7 inhibitor GNE-6776 in non-small cell lung cancer involves regulation of epithelial-mesenchymal transition, cell cycle, Wnt/β-catenin and PI3K/AKT/mTOR pathways" authors investigated the effects and mechanisms of the USP7 inhibitor GNE-6776 on human NSCLC A549 and H1299 cells, providing insights for anti-NSCLC drug development. The authors performed experiments well and it can be considered for publication in this journal after revision.
1. How does the MAPK, PI3K/AKT/mTOR, and Wnt changes in the NSLC development? Mention in the introduction.
2. Make a flow chart of the overall experimental procedure for this study.
3. Make a pathway diagram mentioning upregulated and dysregulated proteins due to NSLC development and their perturbation under treatment with GNE-6776 at optimum concentration.
Author Response

(The authors gave the same response as above.)

Reviewer 3 Report
Comments and Suggestions for Authors
1) Clarify the sentence - The USP family is the largest and most extensively studied category within the DUB family.
2) What is the objective of the findings?
3) Improve the background of investigation.
4) Improve the resolution of figure 2.
5) MMPs are known to significantly influence the invasive- 184 ness of NSCLC. Give reference.
6) The overlapping region shows genes shared across groups, 249 indicating potential common targets or pathways modulated by GNE-6776 in NSCLC. What does it mean?
7) Discussion can be improved.
8) Add the conclusion part.
Comments on the Quality of English LanguageThe English language and sentence could be improved.
Author Response

(The authors gave the same response as above.)

Reviewer 4 Report
Comments and Suggestions for Authors
The paper "USP7 inhibitor GNE-6776 represses the proliferation, invasion, and migration of non-small cell lung cancer by regulating the epithelial-mesenchymal transition and cell cycle" written by Wu et al. is a solid work. It contains a set of well-designed experiments providing a genuine impression of the activity of USP7 inhibitors against NSCLC (more specifically against NSCLC A549 and H1299 cells). The authors made many valuable in vitro and in vivo investigations, using such methods as cell proliferation assays, flow cytometry for apoptosis and cell cycle, mitochondrial membrane potential measurement, migration and invasion assays, Western blot analysis for key pathways (PI3K/AKT/mTOR, Wnt/β-catenin, EMT markers), and mouse xenograft models for tumor growth. I think the work is worth publishing, but I still have some recommendations. First, as it is not possible to measure IC50 values from proliferation assays, I recommend increasing the duration of the experiment and maybe reducing the FBS in the medium to obtain clearer data; the same could work for other essays also (e.g. cell cycle arrest, where results are not that sharply evident even though p < 0.05). Second, it would be good to include positive and negative controls for in vitro assays. Third, toxicity studies with healthy cells (for instance, fibroblasts) are also desirable to confirm the selectivity of the title compound. However, considering good in vivo results, I believe that the paper could be published as it is.
Author Response

(The authors gave the same response as above.)
